# Effect of Speed Sintering on Low Temperature Degradation and Biaxial Flexural Strength of 5Y-TZP Zirconia

**DOI:** 10.3390/molecules27165272

**Published:** 2022-08-18

**Authors:** Suchada Kongkiatkamon, Chaimongkon Peampring

**Affiliations:** Department of Prosthetic Dentistry, Faculty of Dentistry, Prince of Songkla University, Hat Yai, Songkhla 90110, Thailand

**Keywords:** dental ceramics, zirconia, Y-TZP, CAD/CAM, low temperature degradation, flexural strengths, grain size, translucency

## Abstract

Translucent zirconia is becoming the material of choice for the esthetic restorative material. We aimed to evaluate the surface structure, phase determination, translucency, and flexural strength of 5Y-TZP Zirconia (Katana STML Block and Disc) between the regular sintering and the speed sintering with and without low-temperature degradation (LTD). A total of 60 zirconia discs (30 per group; regular sintering and speed sintering) were used in this study. A CAM machine was used to mill cylinders out of the zirconia blanks and then cut into smaller discs. For the speed sintering, the zirconia blocks were milled into smaller discs. The zirconia discs were subjected to regular and speed sintering with and without LTD. Scanning electron microscopy was used to characterize the zirconia specimens and the zirconia grain size. Furthermore, the zirconia specimens were analyzed for elemental analysis using energy dispersive spectroscopy and phase identification using X-ray diffraction. The zirconia specimens were subjected to translucency measurements and biaxial flexural strength testing. The results of the zirconia specimens were compared among the groups. Statistical analysis was completed using SPSS version 20.0 to detect the statistically significant differences (*p* value = 0.05). A one-way ANOVA with multiple comparisons was performed using Scheffe analysis among the groups. The speed sintering presented smaller grain sizes. The zirconia specimens with and without LTD in regular and speed sintering presented a similar surface structure. Regular sintering showed more translucency compared to speed sintering. Multiple comparisons of the translucency parameter were a significant difference (*p* value < 0.05) between the various groups except for the comparison between speed sintering and speed sintering LTD. The regular sintering showed bigger gain sizes and slightly more translucency compared to speed sintering. The speed sintering showed higher biaxial flexural strengths compared to regular sintering. This shows that speed sintering can be considered a suitable method of sintering zirconia.

## 1. Introduction

Recently, there has been a substantial development in restorative biomaterials and technologies in dentistry [1,2,3,4,5]. The all-ceramic/zirconia restorations are esthetic restorations and are widely used restorative materials in restorative and prosthetic dentistry [6]. Zirconia is a polymorphic biomaterial having three crystallographic chemical structures: monoclinic, tetragonal, and cubic [7,8]. Yttria-partially stabilized zirconia (Y-PSZ), with a higher yttria content (4–6 mol% yttria) than 3Y-TZP, has been developed as a novel dental zirconia material [9].

The zirconia is present in the monoclinic form at room temperature and when heated; its phase transformation occurs to tetragonal and further heating leads to its cubic form [10,11,12], as shown in Figure 1. On cooling, cubic–tetragonal transformation takes place.

Furthermore, zirconia (TZP, tetragonal *zirconia* polycrystal) can be of three types according to the yttria content [13,14]: 3Y-TZP, 4Y-TZP, and 5Y-TZP containing 3 mol%, 4 mol%, 5 mol% Y-TZP, respectively. The 3Y-TZP is strong and mainly tetragonal, 4Y-TZP is more translucent, and 5Y-TZP is the most translucent.

High-precision ceramic prostheses can be produced from computer-aided design and computer-aided manufacturing (CAD/CAM) technology with a more time-efficient and simplified process compared to traditional techniques [15,16]. In addition, milling also facilitates the fabrication of durable ceramic prostheses.

The aging or hydrothermal aging or low-temperature degradation (LTD) of zirconia results in the progressive transformation of the tetragonal to the monoclinic (t–m) phase in the presence of water/water vapor (at approx. 30 °C up to 400 °C). Yttria, a dopant used in zirconia to achieve the stability of the phase, is exhausted following LTD through a reaction in the presence of moisture resulting in the phase transformation [12,17,18].

There are some studies that compared the fast and slow sintering protocol of zirconia [18,19,20,21,22]. However, there is no study that studied speed sintering with LTD. Sintering times with aging can affect the structural, optical, and mechanical properties of zirconia [20]. Hence, the objective of this research is to compare the surface structure, phase determination, translucency, and biaxial flexural strength of zirconia blocks and discs (5Y-TZP) between the regular speed sintering and the speed sintering with and without LTD. The null hypothesis is that there is no difference in the surface structure, grain size, contrast ratio, translucency, and flexural strength in zirconia between regular sintering and the speed sintering with and without LTD.

## 2. Materials and Methods

Figure 2 shows the overview of this study. The study design was conducted by dividing into 2 sintering protocols: regular sintering (RS) and speed sintering (SS). A total sample size of 56 was calculated from the G*Power software version 3.1.9.7 [23] with the power of the study at 90%. Hence, 60 zirconia discs (30 per group) were selected for this study.

### 2.1. Specimen Preparation

From the 3rd generation of zirconia (Katana STML Block) (5Y-TZP), specimens were prepared by the CAD/CAM system in both sintering protocols: speed sintering and regular sintering groups (Figure 2 and Figure 3). Thirty zirconia blocks (Katana block 14ZL/STML A3) by the CAD/CAM system (inLAB20) were milled by a CAM machine (MCXL) to produce a disc shape.

For the regular sintering groups, big zirconia discs were milled into a cylinder shape and then cut into smaller discs using a CAM machine (IMES-ICORE Coritec 250i, Germany) (Figure 3A). For the speed sintering, the zirconia blocks (height 17.8 × 19.2 width × length 40 mm) were milled into smaller discs (Figure 3B).

The total thermal cycle, sintering time, and dwell temperature for the regular and speed sintering are shown in Figure 4. For the regular sintering group, the total thermal cycle, sintering time, and dwell temperature for the regular sintering were 6.8 h, 2 h, and 1550 °C, respectively, by using inFire HTC Furnace (Dentsply Sirona, Bensheim, Germany). For the speed sintering group, the total thermal cycle, sintering time, and dwell temperature were calculated based on each specimen, which was 30 min, 16 min, and 1560 °C by using SpeedFire (Dentsply Sirona, Bensheim, Germany).

Dimensions of each zirconia sample were measured with a digital caliper (Mitutoyo; Mitutoyo Co. Ltd., Kawasaki, Japan) before and after sintering. Before sintering, the average size of the zirconia samples was 18.39 mm in diameter and 1.23 mm in thickness. After sintering, the final dimensions for each test were 15 mm in diameter and 1 mm in thickness. The dimensions of the zirconia samples of both sintering protocols were similar.

Then, the specimens were cleaned ultrasonically (Model: 460/M, Elma Schmidbauer GmbH, Singen, Germany) in deionized water for 10 min then air dried individually for 20 s before testing.

### 2.2. Specimen Preparation for Hydrothermal Aging

Specimens for each group (15 for regular sintering and 15 discs for speed sintering) were subjected to LTD using an autoclave machine (TOMY ES 215/ES-315, Tomy Kogyo Co. Ltd., Nerima, Japan) at 122 °C under 2-bar pressure for 8 h and subjected to the phase transformation of zirconia at the surface. A total of 1 h of steam autoclave at 122 °C under 2-bar pressure has the same effect as 1 y in vivo [24]. Each specimen was subjected to sterilization using the autoclave. After the LTD, all specimens were dried in air for 24 h.

### 2.3. Surface Characterization and Phase Structure 

#### 2.3.1. Surface Structure, Grain Size, and Elemental Analysis 

The zirconia specimens and the zirconia grain size were characterized using scanning electron microscopy (SEM) (FEI Quanta 400, FEI, Czech). At first, the zirconia specimens were coated with a gold–palladium alloy using sputter-coating and SEM images were obtained with an accelerating voltage of 20 kV with a 1000×, 5000×, 10,000×, 20,000×, and 30,000× magnification. Grain sizes were determined in each group from the SEM and the average grain size was calculated [20]. For measuring the grain size from the SEM, 3 specimens were randomly selected from each group. The five largest and five smallest grains in the center and off-center of each specimen were measured and the mean sizes were calculated and compared.

Furthermore, the zirconia specimens were analyzed for elemental analysis using energy dispersive spectroscopy (EDS) (OXFORD X-MaxN, FEI, Czech) to quantify material composition (zirconium, yttrium, aluminum, hafnium, and other oxides).

#### 2.3.2. Phase Structure

The phase identification of the zirconia specimens was done using X-ray diffraction (XRD) (Philips X’Pert MPD, Philips, Eindhoven, The Netherlands) using Cu-K_α_ radiation from 0–90° (2ɵ). Scans were performed at 40 kV, 30 mA, step size of 0.05°/step, and a scan time of 1 s/step. The phase structures of zirconia were defined as a tetragonal unit cell with space group P42/nmc, a monoclinic unit cell with space group P21/c, and a cubic with space group Fm-3m (library data). Then, the phase fraction, the peak position, and composition were measured. The fractions volume of the monoclinic transformation was calculated from the peak intensities (Xm). 

### 2.4. Translucency 

The zirconia specimens were subjected to translucency measurements. A spectrophotometer (HunterLab, ColorQuest XE, Hunter Associates Laboratory Inc., Reston, Virginia, USA) with a calibration plate and port size 0.375″ was used to record the CIELAB coordinates (L*, a*, and b*) of the zirconia discs. The translucency parameter (TP) was calculated from the difference in color between the same specimen against black and white backgrounds using the following formula (Equation (1)) [25]:(1)$$\mathrm{TP}=[{\left(L^{*}{\;}_{B}-L^{*}{\;}_{W}\right)}^{2}+{\left(a^{*}{\;}_{B}-a^{*}{\;}_{W}\right)}^{2}+(b^{*}{\;}_{B}-b^{*}{\;}_{W}{)}^{2}{]}^{1/2}$$
where the B and W subscripts are the color coordinates over black and white backgrounds. A higher TP value indicates a higher translucency.

The contrast ratio (CR) was calculated from the spectral reflectance of light (Y) over a black (Y_B_) and white (Y_W_) background using the Equations (2) and (3) [25]:(2)$$Y=\left[(L^{*}+16\right)\;/\;116{]}^{3}\;\times \;100$$
(3)$$\mathrm{CR}=Y_{B}\;/{\;Y}_{W}$$
where the Y = spectral reflectance of light of the specimen over a Y_B_ = black and Y_W_ = white background and CR = contrast ratio. The lower the CR, the more translucent. For a transparent material, CR is 0.0 whereas, for an opaque material, CR is 1.0.

### 2.5. Biaxial Flexural Strength

Zirconia discs (30 per group, Ø = 15 and t = 1 mm) were subjected to the flexural strength tests according to ISO 6872:2015 [26]. The treated surface of the specimens was faced down (tensile stress) on three balls (Ø = 4.6 mm) placed in a triangular position 10 mm apart from each other. A circular tungsten piston (Ø = 1.4 mm) of a universal testing machine (Lloyd Instruments, Model LRX-Plus, AMETEK Lloyd Instrument Ltd., Hampshire, UK) was used with an increasing load (1 mm/min) until the catastrophic failure occurred. Then, the flexural strength was calculated from Equation (4):(4)$$\sigma =[-0.2387P\left(X-Y\right)]\;/{\;b}^{2}$$
where σ is the maximum tensile stress, P is the total load to fracture, and b is the thickness at fracture origin. X and Y are calculated from Equations (5) and (6):(5)$$X=\left(1+\nu \right)\;\mathrm{In}{\left(r_{2}/r_{3}\right)}^{2}+\left[\left(1-\nu \right)/2\;{\left(r_{2}/r_{3}\right)}^{2}\right]$$
(6)$$Y=\left(1+\nu \right)\;{\left[1+\mathrm{In}(r_{1}/r_{3}\right)}^{2}+\left(1-\nu \right)\;{\left(r_{1}/r_{3}\right)}^{2}$$
where ν is Poisson’s ratio (ν = 0.30)^83^, r_1_ is the radius of the support circle (6.05 mm), r_2_ is the radius of the loaded area (0.8 mm), and r_3_ is the radius of the specimen (7.5 mm).

### 2.6. Statistical Analysis

The results of the zirconia specimens were compared among the groups. Descriptive statistics were calculated for each experiment. Statistical analysis was done using SPSS software version 20.0 (SPSS Inc., IBM Corporation, Chicago, IL, USA) to detect statistically significant differences (*p* value = 0.05). One-way ANOVA was used to compare the studied parameters among the groups. Multiple comparisons were done using Scheffe analysis.

## 3. Results

### 3.1. Surface Structure and Elemental Analysis

The SEM images of the zirconia specimens are shown in Figure 5 and the results of the grain size (µm) in each group are shown in Table 1. It showed that regular sintering showed a higher grain size compared to speed sintering. The low-temperature degradation (LTD) zirconia specimens showed a similar structure to no LTD in both regular and speed sintering groups.

Table 2 shows the multiple comparisons of the grain size of the zirconia specimens among various groups among groups. The regular sintering showed a significantly bigger grain size (*p* value < 0.05) compared to the speed sintering LTD and regular sintering LTD compared to the speed sintering and speed sintering LTD. There was no difference in the flexural strength between the regular sintering vs regular sintering LTD (*p* value = 0.975) and speed sintering vs speed sintering LTD (*p* value = 0.995).

The EDS elemental analysis of the zirconia specimens is shown in Figure 6 and Table 3. Figure 6 shows the EDS spectra and Table 3 shows the elemental analysis of the zirconia specimens in atomic %. The speed sintering presented slightly higher Zr. LTD showed slightly lower C and O with higher Y, Zr, and Hf.

### 3.2. Phase Structure

The XRD analysis (Figure 7) of the zirconia specimens shows that the zirconium specimens showed that the peak positions for the spectra correspond to the tetragonal phase for zirconium yttrium oxides (ZrO_2_).

### 3.3. Translucency 

The results of the transparency are shown in Table 4. Regular sintering showed more translucency compared to speed sintering. Multiple comparisons of the contrast ratio between the groups show that there was a significant difference (*p* value < 0.05) between the various groups. However, the translucency parameter presented a significant difference (*p* value < 0.05) between the various groups except for between speed sintering vs speed sintering LTD (*p* value = 0.931).

### 3.4. Biaxial Flexural Strength

The results of the flexural strength (MPa) are shown in Table 4. It showed that speed sintering and speed sintering LTD showed higher biaxial flexural strengths. 

Table 5 shows the multiple comparisons of the flexural strength (MPa) among the groups. The regular sintering showed significantly lower flexural strength (*p* value ≤ 0.01) compared to the speed sintering and speed sintering LTD. Similarly, the regular sintering LTD showed a significantly lower flexural strength (*p* value < 0.005) compared to the speed sintering and speed sintering LTD. However, there was no difference in the flexural strength between the speed sintering and speed sintering LTD (*p* value = 0.444).

## 4. Discussion

Zirconia has good wear resistance and good color stability [27]. There have been developed various sintering methods to enhance the properties and esthetics of zirconia. As there was no study that studied the speed sintering with aging or LTD. Sintering times with LTD aging can affect the structural, optical, and mechanical properties of zirconia [20]. Hence, it was important to study the surface structure, phase determination, translucency, and flexural strength of zirconia between regular sintering and speed sintering. So, this research was done to compare the surface structure, phase determination, translucency, and biaxial flexural strength of zirconia blocks and discs (5Y-TZP) between the regular speed sintering and the speed sintering with and without LTD. In this study, we rejected the null hypothesis as there was a difference in the surface structure, grain size, contrast ratio, and biaxial flexural strength in zirconia between the regular sintering and the speed sintering with and without LTD. In this study, the regular sintering with LTD showed the most translucency. From XRD, there was no difference in all groups. All the peaks were the same; however, the translucencies are different, which could be due to the difference in the grain size. A possible scenario for this change might be due to material structural changes due to the degradation of the metal salts within the coloring liquid [28]. In addition, in this study, for speed sintering, we can notice a smaller grain size. The smaller grain of the monoclinic lattice may cause less light transmission. Liu et al. [9] also found similar results that the conventionally-sintered Y-PSZ had a larger average grain size and a smaller fraction of fine grains than those of the speed-sintered specimens.

Furthermore, Kilinc et al. [20] studied the various sintering methods and aging on the grain size, flexural strength, and translucency of the zirconia and they found that the sintering method and aging significantly influence the translucency. The flexural strength and grain sizes were influenced by aging only (*p* < 0.001). Aging times increased the grain size but prolonged sintering with 120 min of aging negatively influenced the translucency of zirconia. The increase in grain size compared to our research might be due to the shorter aging time (60 min and 120 min), but in our research, the samples were subjected to 122 °C under 2-bar pressure for 8 h. Hence, prolonged aging reduces the grain size. Similarly, Alamledin et al. [29], mentioned that accelerated aging had an effect on the translucency of both fully-stabilized cubic and partially-stabilized tetragonal zirconia, causing an increase in the translucency of zirconia.

In this study, the size of the zirconium samples before sintering was 1.226 times bigger than the size of the zirconium samples after sintering. This bigger size before sintering is to compensate for the shrinkage following the sintering (enlargement factor of 1.226). In addition, the dimensions of the zirconia samples of both sintering protocols were similar. Ahmed et al. [30] also mentioned that there is no difference in the linear and volumetric dimensional changes between standard and fast sintering protocols.

Liu et al. [22] studied the optical properties of two generations of rapid sintered translucent zirconia (3Y-TZP and 5Y-TZP) and they found that rapid sintering resulted in reduced lightness but did not affect the surface roughness. They concluded that rapid sintering is a practical method of reducing the production time of zirconia restorations. 

In our study, the flexural strength was affected by the sintering method. These results could be due to the smaller grain size of zirconia from speed sintering. These results were similar to Ersoy et al. [21], who found that both a high and low sintering temperature combination increases the zirconia’s flexural strength, but there were no visible differences in the grain size in the zirconia specimens. Stawarczyk et al. [31] mentioned that the grain size of zirconia increases with the sintering temperatures and the sintering temperature significantly affects the flexural strength.

When heating or low-temperature degradation (LTD), the monoclinic phase will transform into a tetragonal and cubic phase according to the temperature (Figure 1), and when cooling, it will transform back to monoclinic, which is not strong. Obtaining stable sintered zirconia ceramic products is difficult because of the large volume change accompanying the transition from tetragonal to monoclinic (approximately 5%). Hence, yttria is added to stabilize in the tetragonal phase [32,33]. Yttria-doping reduced the grain growth, stabilized the tetragonal phase, and significantly improved the thermal stability of the particles. In addition, stabilization of the cubic polymorph of zirconia over a wider range of temperatures is accomplished by the substitution of some of the Zr4+ ions (ionic radius of 0.82 Å, too small for the ideal lattice of fluorite characteristics for the cubic zirconia) in the crystal lattice with slightly larger ions, e.g., those of Y3+ (ionic radius of 0.96 Å). The resulting doped zirconia materials are stabilized zirconia (PSZ) [32].

Arcila et al. [34] characterized the microstructure of three types of zirconia; 3Y-TZP (yttria-tetragonal *zirconia* polycrystal), 4Y-PSZ, and 5Y-PSZ (yttria-partially stabilized zirconia) and compared their hardness, fracture resistance, and fatigue and flexural strength. Three zirconia used were 3Y-TZP (Vita YZ HT), 4Y-PSZ (Vita YZ ST), and 5Y-PSZ (Vita YZ XT). The 4Y-PSZ and 5Y-PSZ specimens presented some surface defects under the SEM, whereas the 3Y-TZP demonstrated a greater grain consistency on the surface. They concluded that despite the structural differences, 4Y-PSZ and 3Y-TZP had similar fatigue. The 5Y-PSZ had the least mechanical strengths.

Finally, in this study, only one brand of zirconia was used, and different manufacturers may have presented different grain sizes for sintering zirconia. Future studies can be done to study more the grain size and its chemistry.

## 5. Conclusions

This research showed that there was a difference in surface structure, translucency, and biaxial flexural strengths. The regular sintering showed a bigger grain size and slightly more translucency compared to speed sintering. The speed sintering showed higher biaxial flexural strengths compared to regular sintering. This shows that speed sintering can be considered a suitable method of sintering zirconia. Hence, when biaxial flexural strength is required, speed sintering can be considered; however, when better translucency is required, regular sintering is recommended.

## Figures and Tables

**Figure 1 molecules-27-05272-f001:**
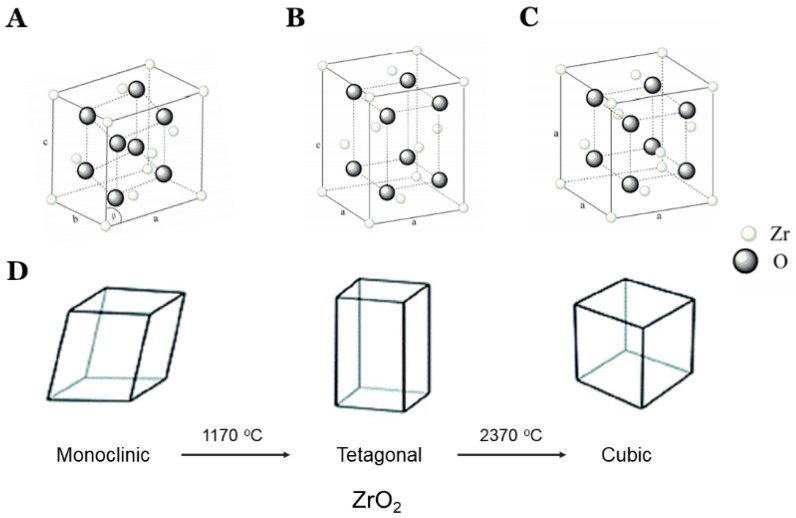
Different phases and phase transformation of zirconia. Monoclinic (**A**), tetragonal (**B**), and cubic structure (**C**). Zirconia can have 3 polymorphs depending on temperature: monoclinic at <1170 °C, tetragonal between 1170 °C and 2370 °C, and cubic at >2370°C (**D**). Adapted with permission from Ref. [11]. 2019, MDPI, Basel, Switzerland.

**Figure 2 molecules-27-05272-f002:**
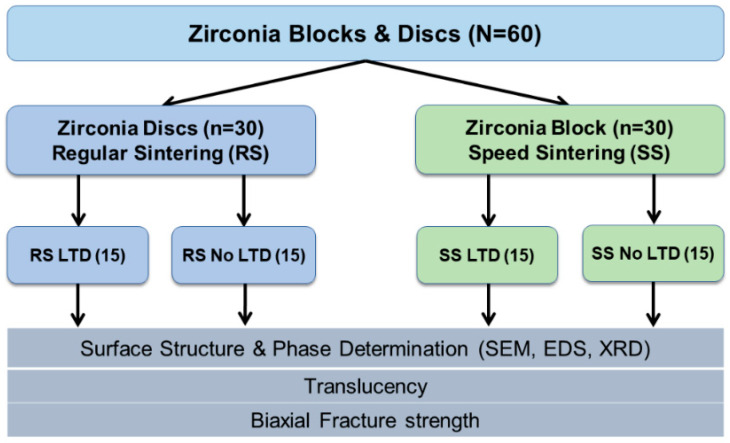
Overview method of the study. LTD = low temperature degradation.

**Figure 3 molecules-27-05272-f003:**
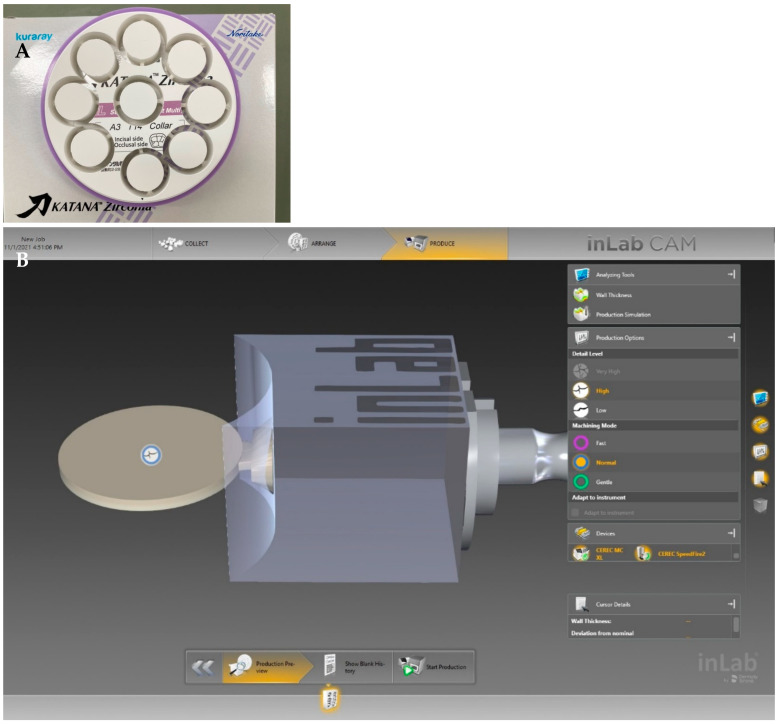
Preparation of the zirconia specimens by milling of big zirconia disc into a cylinder shape and then cutting into smaller discs for the regular sintering (**A**) and milling of the zirconia block into smaller discs for the speed sintering (**B**).

**Figure 4 molecules-27-05272-f004:**
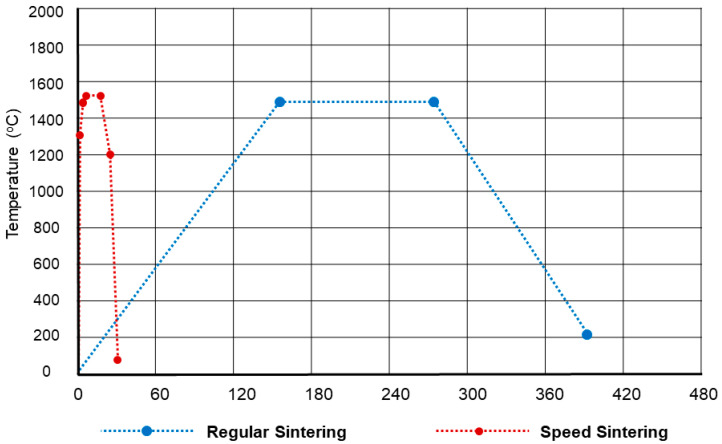
Total thermal cycle, sintering time, and dwell temperature for the regular and speed sintering.

**Figure 5 molecules-27-05272-f005:**
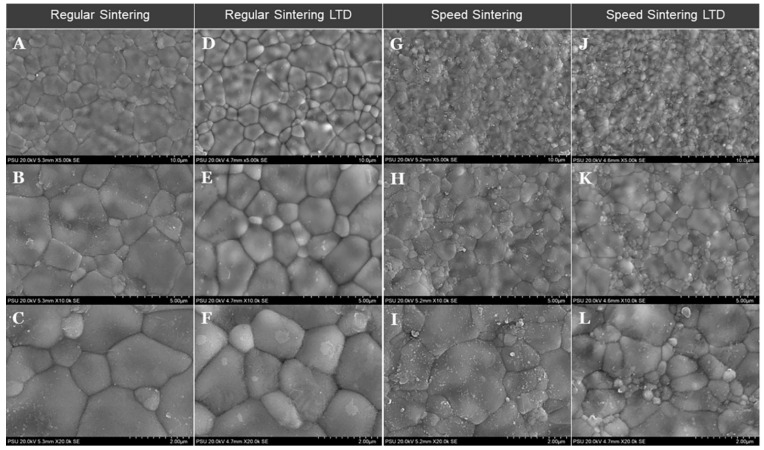
Scanning electron microscopy images of the zirconia specimens for the regular sintering and speed sintering with and without low-temperature degradation (LTD). Regular sintering (**A**–**F**) and speed sintering (**G**–**L**) from low magnification to high magnification.

**Figure 6 molecules-27-05272-f006:**
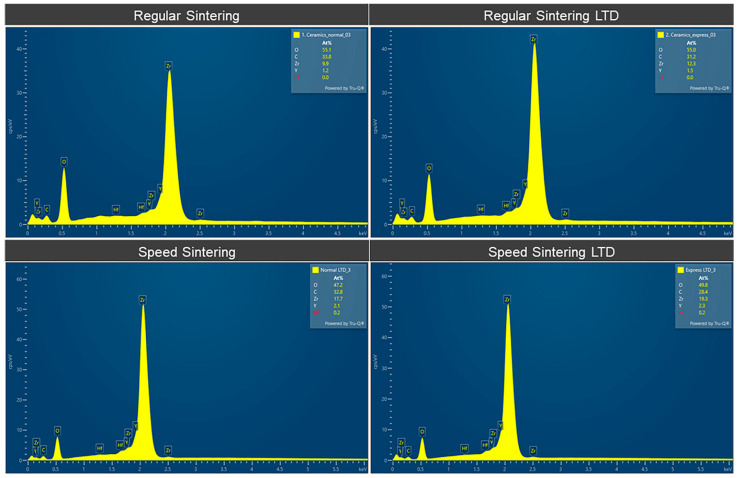
EDS Spectra of the zirconia specimens for the regular speed sintering and speed sintering with and without low-temperature degradation (LTD).

**Figure 7 molecules-27-05272-f007:**
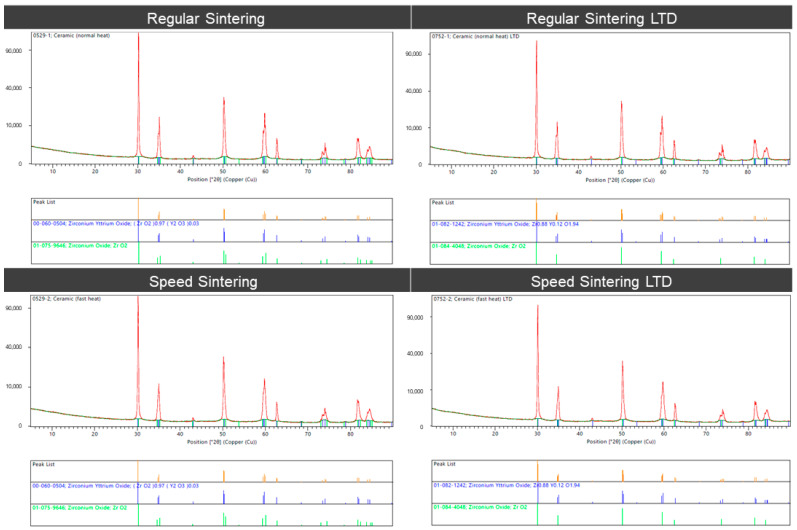
X-ray diffraction (XRD) analysis of the zirconia specimens for the regular sintering and speed sintering with and without low-temperature degradation (LTD).

**Table 1 molecules-27-05272-t001:** Results of the grain sizes of the zirconia specimens in various groups.

Groups	Mean (µm)	SD (µm)
Regular Sintering	3.206	1.257
Regular Sintering LTD	3.442	1.820
Speed Sintering	1.822	0.452
Speed Sintering LTD	1.689	1.053

SD = Standard deviation.

**Table 2 molecules-27-05272-t002:** Multiple comparisons of the grain size of the zirconia specimens among various groups.

(I) Groups	(J) Groups	Mean Difference (I − J)	*p* Value
Regular Sintering	Regular Sintering LTD	−0.236	0.975
	Speed Sintering	1.383	0.075
	Speed Sintering LTD	1.517	0.042 *
Regular Sintering LTD	Regular Sintering	0.236	0.975
	Speed Sintering	1.6192	0.026 *
	Speed Sintering LTD	1.7532	0.014 *
Speed Sintering	Regular Sintering	−1.383	0.075
	Regular Sintering LTD	−1.6192	0.026 *
	Speed Sintering LTD	0.134	0.995
Speed Sintering LTD	Regular Sintering	−1.517	0.042 *
	Regular Sintering LTD	−1.753	0.014 *
	Speed Sintering	−0.134	0.995

* The mean difference is significant at the 0.05 level. Multiple comparisons were done using Scheffe analysis.

**Table 3 molecules-27-05272-t003:** Results of the EDS elemental analysis of the zirconia specimens for the regular speed sintering and speed sintering with and without low-temperature degradation (LTD).

Zirconia Specimens	C (Mean ± SD) (Atomic %)	O (Mean ± SD) (Atomic %)	Y (Mean ± SD) (Atomic %)	Zr (Mean ± SD) (Atomic %)	Hf (Mean ± SD) (Atomic %)
Regular Sintering	35.073 ± 1.355	53.773 ± 1.505	1.220 ± 0.051	9.833 ± 0.238	0.016 ± 0.004
Regular Sintering LTD	32.973 ± 1.682	47.436 ± 0.947	2.076 ± 0.084	17. 376 ± 0.6930	0.1433 ± 0.024
Speed Sintering	34.260 ± 2.692	53.123 ± 1.831	1.346 ± 0.097	11.163 ± 0.906	0.013 ± 0.004
Speed Sintering LTD	28.190 ± 0.190	50.093 ± 0.339	2.316 ± 0.02	19.236 ± 0.182	0.166 ± 0.012

SD = Standard deviation.

**Table 4 molecules-27-05272-t004:** Results of the contrast ratios transparency, and biaxial flexural strength of the zirconia specimens.

Zirconia Specimens	Contrast Ratio (Mean ± SD)	Translucency (Mean ± SD)	Flexural Strength (MPa) (Mean ± SD)
Regular Sintering	0.787 ± 0.034	10.581 ± 0.798	466.41 ± 22.898
Regular Sintering LTD	0.734 ± 0.028	12.443 ± 1.173	471.85 ± 20.789
Speed Sintering	0.833 ± 0.021	9.052 ± 0.618	500.5 ± 23.432
Speed Sintering LTD	0.824 ± 0.252	9.263 ± 0.775	513.63 ± 19.909

SD = Standard deviation.

**Table 5 molecules-27-05272-t005:** Multiple comparisons of the biaxial flexural strength of the zirconia specimens among various groups.

(I) Groups	(J) Groups	Mean Difference (I − J)	*p* Value
Regular Sintering	Regular Sintering LTD	−5.436	0.926
	Speed Sintering	−34.097 *	0.001 *
	Speed Sintering LTD	−47.219 *	<0.0001 *
Regular Sintering LTD	Regular Sintering	5.436	0.926
	Speed Sintering	−28.661 *	0.008 *
	Speed Sintering LTD	−41.783 *	<0.0001 *
Speed Sintering	Regular Sintering	34.097 *	0.001 *
	Regular Sintering LTD	28.661 *	0.008 *
	Speed Sintering LTD	−13.122	0.444
Speed Sintering LTD	Regular Sintering	47.219 *	<0.0001 *
	Regular Sintering LTD	41.783 *	<0.0001 *
	Speed Sintering	13.122	0.444

* The mean difference is significant at the 0.05 level. Multiple comparisons were done using Scheffe analysis.

## Data Availability

The data presented in this study are available on request from the corresponding authors.

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
