# Peer review of "Effect of Speed Sintering on Low Temperature Degradation and Biaxial Flexural Strength of 5Y-TZP Zirconia"

_molecules, 2022, doi:10.3390/molecules27165272_

Round 1
Reviewer 1 Report
The present work studies the effect of conventional vs. speed sintering as also that of low temperature degradation on 5Y-TZP for biomaterial applications. Translucency and flexural strength are the major properties of concern in the materials. The work tries to correlate these properties with the microstructure of the thermally treated disc samples. A large number of samples are used to ascertain the statistical significance of the obtained results. However, while the reporting of the flexural strength data is rigorous, the microstructural data is not. This in my view has led to incorrect observations and conclusions. At the same time, it is not clear exactly what the novelty of this work is in the absence of sufficient prior work cited in the introduction section and how this work fits in. In view of this, I cannot recommend publication of this manuscript unless significant improvements and corrections are made.
To qualify the above summary, I have detailed major and minor comments below.
Major comments:
1. Title: There are no fatigue measurements anywhere in the paper. The title is misleading.
2. Line 11 – 30: This section of the abstract pretty much describes the procedure and equipment used for the work and as such should be included in the appropriate section. Please re-write the abstract to make it more concise and direct.
3. Line 61-63: Please include the reaction through which yttrium is “exhausted” from the zirconia during low temperature thermal degradation.
4. Line 68: The statement about there being no prior work is not necessarily true. Although there might not be a single work which repeats these exact conditions, there is significant similar work in the field which looks at similar properties and applications. This is also evidenced by the authors citing a number of prior works in the discussion section. It is pertinent that authors detail how the current work is different from the existing work and explicitly state what new knowledge is generated or expected to be generated.
5. Line 91-92: What exactly do the authors mean by a ‘block’? What are its dimensions? When the samples were milled from the block, what was their shape and size? Also, why were different shaped samples used for the two different sintering protocols used in the work? Please clarify.
6. Line 100-101: Please also include the ramp up and cool down rates since those can have a significant effect on the microstructure of the zirconia. It will also be useful to get a visual comparison of the sintering profiles in terms of a temperature vs time graph.
7. Line 104-106: Are the dimensions mentioned for unsintered samples? How did the different sintering cycles affect the final dimensions? Also were there any changes in dimensions due to LTD? Please clarify.
8. Line 123: The authors mention that the results are mentioned in the form of average grain sizes. However, no numerical data regarding this is reported anywhere in the text. Since this grain size is a crucial part of the argument being made, please include numerical average grain sizes for the different conditions calculated as per ASTM standards along with appropriate standard errors.
9. Line 175-176: The statement says that the speed sintered samples show a smaller grain size than regularly sintered samples. This statement is factually incorrect as the microstructures in figure show exactly the opposite where the speed sintered samples seem to show a significantly higher grain size. This is where the numerical values calculated per ASTM standards are required. If this statement is incorrect then any structure property relationships built based on it as relating to the flexural strength and transparency are also incorrect. Please correct accordingly.
10. Line 176-177: I would argue that the statement about LTD and no LTD samples showing similar microstructures is also not completely correct. It is very evident from the regular sintering samples, especially when viewed at higher magnifications, that the no LTD samples have a generally higher grain size. Please correct appropriately.
11. Figure 3. Please include figure sub numbers for each number and include the description in the figure caption. It also seems like the regular sintered samples have a distinctly bimodal distribution of grain sizes. Could the authors please comment on that and how it might tie into the mechanical and optical properties measured?
12. Line 182-183: EDS data does not provide any information about grain sizes. Please remove the statement claiming smaller grain sizes for speed sintering.
13. Figure 4: It is not clear where the carbon depicted in the EDS analysis coming from. The samples were coated with gold-palladium to mitigate charging, so no carbon is expected. The authors are requested to clarify and correct appropriately. In the tables below the images noting the elemental composition, please explicitly mention that the percentages are wt%. This is not easy to identify from the images. Also, it will perhaps be more helpful to report at% instead of wt% to better correlate with thermal degradation reactions and expected products.
14. Line 134-135: No monoclinic transformations are reported from the XRD data in the results section. Please remove this sentence.
15. Line 257-258: The qualitative microstructural analysis seems to have drawn wrong conclusions and suffers because of the absence of a quantitative characterization of the grain size.
Minor comments:
1. Line 17: Consider using past tense in the line ‘For the speed sintering, the zirconia block will be milled into smaller discs’.
2. Line 35-36: Awkward phrasing. Please rephrase the sentence.
3. Line 41: ‘There have been substantial’ instead of ‘there is substantial’
4. Line 42-44: Awkward sentence construction. Please consider rephrasing.
5. Line 41-55: There is no continuity between the first and second paragraph of the ‘Introduction’ section. Please ensure a continuous narrative that can be easily followed.
6. Line 48: Use the word ‘three’ instead of the number ‘3’.
7. Line 52-54: The sentence mentions 3Y-TZP thrice instead of 3Y-TZP, 4Y-TZP and 5Y-TZP. Please correct appropriately. Also, zirconia being only of three types according to yttrium content is a misleading statement. Please clarify that these yttrium dopant contents apply only in the case of tetragonal zirconia polycrystal. Higher concentrations of yttrium dopant are certainly possible eg. 8 mol% yttria doped zirconia which gets stabilized in the cubic phase. At the same time, non-whole-number dopant concentrations are also possible. Perhaps rephrase to indicate that the three grades mentioned are more popular in the field of biomaterials.
8. Line 53: Please define the abbreviations TZP, 3Y-TZP and in the subsequent sentence also 4Y-TZP and 5Y-TZP.
9. Line 77: Use the word ‘two’ instead of the number ‘2’.
10. Line 79-80: There is no reference in the list of references for citation 31. Same goes for 32, 33, 34 and 35. These citations are also not in the correct numerical order in the text.
11. Line 87: Please capitalize ‘cam’.
12. Line 91-92: Please use past tense.
13. Line 116: Please correct ‘on air’ to ‘in air’.
14. Lines 142, 148, 160, 163: Please give equation numbers to all the equations and cross reference appropriately in the text.
15. Table 2: Instead of ‘normal’ sintering, please use ‘regular’ sintering to maintain consistency with the rest of the paper. It will be helpful to visualize this data in graphs.
16. Line 228-229: Again, this statement about smaller grain size for speed sintered samples is not corroborated by any data presented.
17. Line 251: Please include a definition of the abbreviation PSZ.
Author Response
Response to Reviewer 1 Comments
The present work studies the effect of conventional vs. speed sintering as also that of low temperature degradation on 5Y-TZP for biomaterial applications. Translucency and flexural strength are the major properties of concern in the materials. The work tries to correlate these properties with the microstructure of the thermally treated disc samples. A large number of samples are used to ascertain the statistical significance of the obtained results. However, while the reporting of the flexural strength data is rigorous, the microstructural data is not. This in my view has led to incorrect observations and conclusions. At the same time, it is not clear exactly what the novelty of this work is in the absence of sufficient prior work cited in the introduction section and how this work fits in. In view of this, I cannot recommend publication of this manuscript unless significant improvements and corrections are made. To qualify the above summary, I have detailed major and minor comments below.
Thank you for your positive comments. Corrections in the Manuscript for Reviewer 1 are highlighted in Yellow color.
Major comments:
- Title: There are no fatigue measurements anywhere in the paper. The title is misleading.
Response: The title is improved.
- Line 11 – 30: This section of the abstract pretty much describes the procedure and equipment used for the work and as such should be included in the appropriate section. Please re-write the abstract to make it more concise and direct.
Response: Abstract is edited and improved.
- Line 61-63: Please include the reaction through which yttrium is “exhausted” from the zirconia during low temperature thermal degradation.
Response: When heating or low-temperature degradation (LTD), the monoclinic phase will transform into tetragonal and cubic phase according to the temperature. And then cooling, it will transform back to monoclinic which is not strong. Hence, yttria are added to stabilized in tetragonal phase. Yttria-doping reduced grain growth, stabilized the tetragonal phase, and significantly improved the thermal stability of the particles. (Figure 6).
- Line 68: The statement about there being no prior work is not necessarily true. Although there might not be a single work which repeats these exact conditions, there is significant similar work in the field which looks at similar properties and applications. This is also evidenced by the authors citing a number of prior works in the discussion section. It is pertinent that authors detail how the current work is different from the existing work and explicitly state what new knowledge is generated or expected to be generated.
Response: Although, there are some studies that studied the speed sintering protocol, there is no study that studied at the speed sintering with LTD. Details are added in the introduction. (Page 2).
- Line 91-92: What exactly do the authors mean by a ‘block’? What are its dimensions? When the samples were milled from the block, what was their shape and size? Also, why were different shaped samples used for the two different sintering protocols used in the work? Please clarify.
Response: For the speed sintering, the zirconia blocks (height 17.8 x 19.2 width x length 40 mm) were milled into smaller discs. Blocks were small so big disks were used to produce small disks were used for the regular sintering. (Page 3)
From the 3rd generation of zirconia (Katana STML Block) (5Y-TZP) specimens were prepared by the CAD/CAM system (Figure 2). For the regular sintering groups, big zirconia discs were milled into a cylinder shape and then cut into smaller discs using a CAM machine (IMES-ICORE Coritec 250i, Germany) (Figure 2A). For the speed sintering, the zirconia blocks were milled into smaller discs (Figure 2B). Figure 2 is edited for better clarification.
- Line 100-101: Please also include the ramp up and cool down rates since those can have a significant effect on the microstructure of the zirconia. It will also be useful to get a visual comparison of the sintering profiles in terms of a temperature vs time graph.
Response: Added Figure 4.
- Line 104-106: Are the dimensions mentioned for unsintered samples? How did the different sintering cycles affect the final dimensions? Also were there any changes in dimensions due to LTD? Please clarify.
Response: The dimensions of the specimens before and after sintering are added in the Page 4 and 10.
Discussion: In this study, the size of the zirconium samples before sintering were 1.226 times bigger than size of the zirconium samples after sintering. This bigger size before sintering is to compensate the shrinkage following the sintering (enlargement factor of 1.226). In addition, the dimensions of the zirconia samples of both sintering protocols were similar. Ahmed et al. also mentioned that there is no difference in the linear and volumetric dimensional changes between standard and fast sintering protocols.
- Line 123: The authors mention that the results are mentioned in the form of average grain sizes. However, no numerical data regarding this is reported anywhere in the text. Since this grain size is a crucial part of the argument being made, please include numerical average grain sizes for the different conditions calculated as per ASTM standards along with appropriate standard errors.
Response: We did further test to measure the quantitative analysis of the grains size from the SEM (Table 1 and 2).
Table 2 shows the multiple comparisons of the grains size of the zirconia specimens among various groups among the groups. The regular sintering showed significantly bigger grains size (P value <0.05) compared to the speed sintering LTD and regular sintering LTD compared to the speed sintering and speed sintering LTD. There was no difference in the flexural strength between the regular sintering vs regular sintering LTD (P value = 0.975) and speed sintering vs speed sintering LTD (P value = 0.995).
- Line 175-176: The statement says that the speed sintered samples show a smaller grain size than regularly sintered samples. This statement is factually incorrect as the microstructures in figure show exactly the opposite where the speed sintered samples seem to show a significantly higher grain size. This is where the numerical values calculated per ASTM standards are required. If this statement is incorrect then any structure property relationships built based on it as relating to the flexural strength and transparency are also incorrect. Please correct accordingly.
Response: We did further test to measure the quantitative analysis of the grains size from the SEM (Table 1 and 2) and correct Figures are added.
- Line 176-177: I would argue that the statement about LTD and no LTD samples showing similar microstructures is also not completely correct. It is very evident from the regular sintering samples, especially when viewed at higher magnifications, that the no LTD samples have a generally higher grain size. Please correct appropriately.
Response: There was error in the SEM Figures. We confirmed with the technician and the correct figures are corrected. We did further test to measure the quantitative analysis of the grains size from the SEM (Table 1 and 2).
- Figure 3. Please include figure sub numbers for each number and include the description in the figure caption. It also seems like the regular sintered samples have a distinctly bimodal distribution of grain sizes. Could the authors please comment on that and how it might tie into the mechanical and optical properties measured?
Response: Figure sub numberings are added.
We did further test to measure the quantitative analysis of the grains size from the SEM (Table 1 and 2) and discussed.
- Line 182-183: EDS data does not provide any information about grain sizes. Please remove the statement claiming smaller grain sizes for speed sintering.
Response: The sentence is removed.
- Figure 4: It is not clear where the carbon depicted in the EDS analysis coming from. The samples were coated with gold-palladium to mitigate charging, so no carbon is expected. The authors are requested to clarify and correct appropriately. In the tables below the images noting the elemental composition, please explicitly mention that the percentages are wt%. This is not easy to identify from the images. Also, it will perhaps be more helpful to report at% instead of wt% to better correlate with thermal degradation reactions and expected products.
Response: EDS analyses are done again, and the elemental analyses are added in the atomic %. Elemental analysis results are separated from the Figure and separate Table (Table 1) is added for better visualization.
- Line 134-135: No monoclinic transformations are reported from the XRD data in the results section. Please remove this sentence.
Response: The line is removed.
- Line 257-258: The qualitative microstructural analysis seems to have drawn wrong conclusions and suffers because of the absence of a quantitative characterization of the grain size.
Response: We did further test to measure the quantitative analysis of the grains size from the SEM (Table 1 and 2) and discussed.
Minor comments:
- Line 17: Consider using past tense in the line ‘For the speed sintering, the zirconia block will be milled into smaller discs’.
Response: Corrected.
- Line 35-36: Awkward phrasing. Please rephrase the sentence.
Response: The crowns were
- Line 41: ‘There have been substantial’ instead of ‘there is substantial’
Response: Corrected.
- Line 42-44: Awkward sentence construction. Please consider rephrasing.
Response: Improved.
- Line 41-55: There is no continuity between the first and second paragraph of the ‘Introduction’ section. Please ensure a continuous narrative that can be easily followed.
Response: The introduction is edited.
- Line 48: Use the word ‘three’ instead of the number ‘3’.
Response: ‘3’ is replaced with ‘three’. (Page 2, 5, 10)
- Line 52-54: The sentence mentions 3Y-TZP thrice instead of 3Y-TZP, 4Y-TZP and 5Y-TZP. Please correct appropriately. Also, zirconia being only of three types according to yttrium content is a misleading statement. Please clarify that these yttrium dopant contents apply only in the case of tetragonal zirconia polycrystal. Higher concentrations of yttrium dopant are certainly possible eg. 8 mol% yttria doped zirconia which gets stabilized in the cubic phase. At the same time, non-whole-number dopant concentrations are also possible. Perhaps rephrase to indicate that the three grades mentioned are more popular in the field of biomaterials.
Response: These lines are corrected and rephrased.
- Line 53: Please define the abbreviations TZP, 3Y-TZP and in the subsequent sentence also 4Y-TZP and 5Y-TZP.
Response: Abbreviations are defined. (Page 10)
- Line 77: Use the word ‘two’ instead of the number ‘2’.
Response: The crowns were. (Page 2)
- Line 79-80: There is no reference in the list of references for citation 31. Same goes for 32, 33, 34 and 35. These citations are also not in the correct numerical order in the text.
Response: Previously there was error in the referencing. Now, the references are corrected.
- Line 87: Please capitalize ‘cam’.
Response: ‘cam’ is capitalized. (Page 3)
- Line 91-92: Please use past tense.
Response: Corrected.
- Line 116: Please correct ‘on air’ to ‘in air’.
Response: Corrected. (Page 4)
- Lines 142, 148, 160, 163: Please give equation numbers to all the equations and cross reference appropriately in the text.
Response: Equations are numbered and cross reference done.
- Table 2: Instead of ‘normal’ sintering, please use ‘regular’ sintering to maintain consistency with the rest of the paper. It will be helpful to visualize this data in graphs.
Response: ‘Normal’ sintering is changed to ‘regular’ sintering throughout the manuscript.
- Line 228-229: Again, this statement about smaller grain size for speed sintered samples is not corroborated by any data presented.
Response: We did further test to measure the quantitative analysis of the grains size from the SEM (Table 1 and 2).
- Line 251: Please include a definition of the abbreviation PSZ.
Response: PSZ is partially stabilized zirconia. Added in the Page 10.
Reviewer 2 Report
I recommended that the conclusion part would be improved. So, i felt that too short
The conclusion is the part of the research paper that brings everything together in a logical manner. As the last part of a research paper, a conclusion provides a clear interpretation of the results of your research in a way that stresses the significance of your study.
Author Response
Response to Reviewer 2 Comments
Thank you for your positive comments. Corrections in the Manuscript for Reviewer 1 are highlighted in Turquoise color.
I recommended that the conclusion part would be improved. So, I felt that too short.
The conclusion is the part of the research paper that brings everything together in a logical manner. As the last part of a research paper, a conclusion provides a clear interpretation of the results of your research in a way that stresses the significance of your study.
Response: The conclusion is improved.
Reviewer 3 Report
This study aims to reveal the effect of the sintering speed for 5Y-TZP on their optical, mechanical properties, and crystal structure. The results indicated that the regular sintering speed gave a more transparent zirconia while the speed sintering showed higher flexural strength. The result is of interest in the dental materials field. However, further study should be performed before publication. The comments are listed as follows.
1. Title
The title should be changed because fatigue resistance of the zirconia was not examined in this study. The title includes your focus or main findings.
2. Mistake
In line 53, “3Y-TZP, 3Y-TZP, and 3Y-TZP containing 3 mole %, 3 mole %, 3 mole % Y-TZP”
Please recheck them.
3. Add reference
In line 64, “Various studies are done to compare the fast and slow sintering of zirconia”
However, only one reference has been referred. Please add appropriate references.
4. The reason for 5Y-TZP
Please mention why 5Y-TZP was chosen in this study.
5. Sintering speed
Why did you choose the present sintering speed? The manufacturer’s recommendation? Please clarify the reason.
6. Surface finishing
Did you perform surface polishing for the zirconia sample? The roughness of the sample would affect flexural strength.
7. EDS
In Figure 4, the numerical value for each element was inserted. This is At%? Please add the unit.
8. Grain size
In lines 175-176, “It showed that the speed sintering presented smaller grain size”. However, no data were presented on the grain size. We cannot estimate the grain size from the present SEM image. I recommended further experiment to evaluate the grain size quantitively.
9. Effect of the sintering speed on the grain size
Please discuss the effect.
10. Table 2
The results were duplicated. For instance, “Normal sintering vs. Normal sintering LTD” and “Normal sintering LTD vs. Normal sintering” is the same. Please remove either one.
11. Effect of grain size on translucency
In lines 229-230, “The smaller grains of the monoclinic lattice may cause less light transmission.” Is it light? Please refer appropriate study and discuss more.
12. Effect of grain size on flexural strength
Why small grains give higher flexural strength? I guess the large grain exhibits higher flexural strength. Please refer appropriate study and discuss more.
13. Quantitative analysis for grain size
In lines 257-259, “no quantitative analysis was done ~ future studies can be done ~”
In this study, however, grain size is the most important factor because it affect the properties. I recommended quantitative analysis for grain size.
14. Novelty
What is the novelty and finding of this study? The findings have been already reported in some previous studies.
Author Response
Response to Reviewer 3 Comments
This study aims to reveal the effect of the sintering speed for 5Y-TZP on their optical, mechanical properties, and crystal structure. The results indicated that the regular sintering speed gave a more transparent zirconia while the speed sintering showed higher flexural strength. The result is of interest in the dental materials field. However, further study should be performed before publication. The comments are listed as follows.
Thank you for your positive comments. Corrections in the Manuscript for Reviewer 1 are highlighted in Green color.
- Title
The title should be changed because fatigue resistance of the zirconia was not examined in this study. The title includes your focus or main findings.
Response: The title is corrected.
- Mistake
In line 53, “3Y-TZP, 3Y-TZP, and 3Y-TZP containing 3 mole %, 3 mole %, 3 mole % Y-TZP”
Please recheck them.
Response: Mole % are corrected.
- Add reference
In line 64, “Various studies are done to compare the fast and slow sintering of zirconia”
However, only one reference has been referred. Please add appropriate references.
Response: Appropriate references are added.
- The reason for 5Y-TZP. Please mention why 5Y-TZP was chosen in this study.
Response: In this study, we chose 5Y-TZP due to its popularity of usage. 5Y-TZP provide esthetic, and strength and it is used more and more at present.
- Sintering speed
Why did you choose the present sintering speed? The manufacturer’s recommendation? Please clarify the reason.
Response: In this study, we compared between the normal and speed sintering. Speed sintering was originally introduced by Dentsply Sirona company.
Speedfire was utilized widely for one visit restoration in order to produce the restoration in the short period of time.
Katana STML block was recommended to be used both for regular sintering protocol and speed sintering protocol. However, the authors were curious if there is any different in biaxial strength and translucency outcome.
- Surface finishing
Did you perform surface polishing for the zirconia sample? The roughness of the sample would affect flexural strength.
Response: We polished the samples before sintering and not after sintering. The sintered specimens were subjected to the flexural strength testing.
Although, there were literatures suggested that polishing zirconia would increases the biaxial strength. However, all specimens were treated in the same conditions.
- EDS
In Figure 4, the numerical value for each element was inserted. This is At%? Please add the unit.
Response: EDS analyses are done again, and the elemental analyses are added in the atomic %. Elemental analysis results are separated from the Figure and separate Table (Table 1) is added for better visualization.
- Grain size
In lines 175-176, “It showed that the speed sintering presented smaller grain size”. However, no data were presented on the grain size. We cannot estimate the grain size from the present SEM image. I recommended further experiment to evaluate the grain size quantitively.
Response: We did further test to measure the quantitative analysis of the grains size from the SEM (Table 1 and 2).
Table 2 shows the multiple comparisons of the grains size of the zirconia specimens among various groups among the groups. The regular sintering showed significantly bigger grains size (P value <0.05) compared to the speed sintering LTD and regular sintering LTD compared to the speed sintering and speed sintering LTD. There was no difference in the flexural strength between the regular sintering vs regular sintering LTD (P value = 0.975) and speed sintering vs speed sintering LTD (P value = 0.995).
- Effect of the sintering speed on the grain size. Please discuss the effect.
Response: We did further test to measure the quantitative analysis of the grains size from the SEM (Table 1 and 2) and discussed in the Discussion (Page 10 and 11).
- Table 2
The results were duplicated. For instance, “Normal sintering vs. Normal sintering LTD” and “Normal sintering LTD vs. Normal sintering” is the same. Please remove either one.
Response: Normal Sintering and Normal Sintering LTD are not same. Normal Sintering LTD means normal sintering with low temperature degradation.
- Effect of grain size on translucency
In lines 229-230, “The smaller grains of the monoclinic lattice may cause less light transmission.” Is it light? Please refer appropriate study and discuss more.
Response: Effect of grain size on translucency is added in the Discussion (Page 11).
- Effect of grain size on flexural strength
Why small grains give higher flexural strength? I guess the large grain exhibits higher flexural strength. Please refer appropriate study and discuss more.
Response: Effect of grain size on flexural strength is added in the Discussion (Page 11).
- Quantitative analysis for grain size
In lines 257-259, “no quantitative analysis was done ~ future studies can be done ~”
In this study, however, grain size is the most important factor because it affects the properties. I recommended quantitative analysis for grain size.
Response: We did further test to measure the quantitative analysis of the grains size from the SEM (Table 1 and 2).
- Novelty
What is the novelty and finding of this study? The findings have been already reported in some previous studies.
Response: Added in the beginning of the discussion.
Round 2
Reviewer 1 Report
The authors have addressed majority of my comments satisfactorily. However, there are still a few small issues with the manuscript, mainly minor spell check related. Some are listed below.
1. 'Grin' instead of grain in caption for table 1.
2. Missing period after 'vs' in line 19 in the abstract.
3. Section 3.3 sentence 2 has red font.
4. Section 4 discussion at the end still mentions that no quantitative analysis was done on grain sizes when there clearly is a quantitative analysis in the paper.
Author Response
Response to Reviewer 1 (Round 2) Comments
The authors have addressed majority of my comments satisfactorily. However, there are still a few small issues with the manuscript, mainly minor spell check related. Some are listed below.
Thank you for your positive comments. Corrections in the Manuscript for Reviewer 1 are highlighted in Yellow color.
- 'Grin' instead of grain in caption for table 1.
Response: It is corrected.
- Missing period after 'vs' in line 19 in the abstract.
Response: It is corrected.
- Section 3.3 sentence 2 has red font.
Response: It is corrected.
- Section 4 discussion at the end still mentions that no quantitative analysis was done on grain sizes when there clearly is a quantitative analysis in the paper.
Response: We agree with the reviewer. This line is removed.
Reviewer 3 Report
The revised manuscript have been improved. However, further improvement should be done before acceptance.
1. Estimation of grain size
In materials and methods section, the reference has been inserted. In order to clarify the estimation and to understand easily for a reader, please explain the method in the present manuscript. Especially, numbers of the samples and observation points in each sample should be clearly stated in the text. As you know, SEM observation sometimes can deceive data.
2. Fig. 8
The scheme in the Fig is common knowledge for dental zirconia. The figure should be eliminated from the manuscript.
3. Crystalline phase
For XRD results, there is no difference between the speed-sintered sample and normal-sintered sample. However, the author discussed about that the transparence difference is due to the crystalline phase. This cannot be acceptable. The author should be re-considered. Please discuss the difference between the speed-sintered sample and normal-sintered sample, based on your obtained results. The author should not be ignored your results.
Author Response
Response to Reviewer 3 (Round 2) Comments
The revised manuscript has been improved. However, further improvement should be done before acceptance.
Thank you for your positive comments. Corrections in the Manuscript for Reviewer 1 are highlighted in Green color.
- Estimation of grain size
In materials and methods section, the reference has been inserted. In order to clarify the estimation and to understand easily for a reader, please explain the method in the present manuscript. Especially, numbers of the samples and observation points in each sample should be clearly stated in the text. As you know, SEM observation sometimes can deceive data.
Response: For measuring the grain size from the SEM, 3 specimens were randomly selected from each group. The five largest and five smallest grains in the center and off-center of each specimen were measured and the mean sizes were calculated and compared.
- Fig. 8
The scheme in the Fig is common knowledge for dental zirconia. The figure should be eliminated from the manuscript.
Response: Previously, Figure 8 was added as a recommendation from the previous reviewer. It is eliminated from the manuscript as a recommendation.
- Crystalline phase
For XRD results, there is no difference between the speed-sintered sample and normal-sintered sample. However, the author discussed about that the transparence difference is due to the crystalline phase. This cannot be acceptable. The author should be re-considered. Please discuss the difference between the speed-sintered sample and normal-sintered sample, based on your obtained results. The author should not be ignored your results.
Response: This statement is edited. From XRD, there was no difference in all groups. All the peaks were the same, however, the translucencies are different which could be due to the difference in the grain size. This is elaborated in the Discussion.
Round 3
Reviewer 1 Report
The authors have satisfactorily addressed all the comments I had.
Reviewer 3 Report
The revised manuscript has been improved to accept for the Journal.
I appreciate your lot of effort.